# Enhancer additivity and non-additivity are determined by enhancer strength in the *Drosophila* embryo

Jacques P Bothma[1†], Hernan G Garcia[2†‡], Samuel Ng[1], Michael W Perry[1§], Thomas Gregor[2,3]*, Michael Levine[1*¶]

[1]Department of Molecular and Cell Biology, University of California, Berkeley, Berkeley, United States; [2]Department of Physics, Princeton University, Princeton, United States; [3]Lewis-Sigler Institute for Integrative Genomics, Princeton University, Princeton, United States

*For correspondence: tg2@ princeton.edu (TG); mlevine@ berkeley.edu (ML)

[†]These authors contributed equally to this work

Present address: [‡]Department of Molecular and Cell Biology, Department of Physics, and Biophysics Graduate Group, University of California, Berkeley, Berkeley, United States; [§]Department of Biology, New York University, New York, United States; [¶]Department of Molecular Biology, and Lewis-Sigler Institute for Integrative Genomics, Princeton University, Princeton, United States

Competing interests: The authors declare that no competing interests exist.

**Abstract** Metazoan genes are embedded in a rich milieu of regulatory information that often includes multiple enhancers possessing overlapping activities. In this study, we employ quantitative live imaging methods to assess the function of pairs of primary and shadow enhancers in the regulation of key patterning genes-*knirps*, *hunchback*, and *snail*-in developing *Drosophila* embryos. The knirps enhancers exhibit additive, sometimes even super-additive activities, consistent with classical gene fusion studies. In contrast, the *hunchback* enhancers function sub-additively in anterior regions containing saturating levels of the Bicoid activator, but function additively in regions where there are diminishing levels of the Bicoid gradient. Strikingly sub-additive behavior is also observed for *snail*, whereby removal of the proximal enhancer causes a significant increase in gene expression. Quantitative modeling of enhancer–promoter interactions suggests that weakly active enhancers function additively while strong enhancers behave sub-additively due to competition with the target promoter.

## Introduction

There is emerging evidence that metazoan genes occur in a complex regulatory landscape encompassing numerous enhancers (*Jeong et al., 2006*; *Hong et al., 2008*; *Frankel et al., 2010*; *Perry et al., 2011*; *Rada-Iglesias et al., 2011*; *Buecker and Wysocka, 2012*; *Lagha et al., 2012*; *Levine et al., 2014*). For example, the mouse *Sonic Hedgehog* gene is regulated by at least 20 different enhancers scattered over a distance of ~1 Mb (*Jeong et al., 2006*). Individual enhancers mediate expression in a variety of different tissues, including the brain, floorplate, and limb buds. Multiple enhancers with overlapping regulatory activities are also used to control gene expression within individual cell types. For example, the transcriptional activation of the gap gene *hunchback (hb)* in the early *Drosophila* embryo is mediated by both a proximal enhancer and distal 'shadow' enhancer that independently mediate activation in response to high levels of the Bicoid activator gradient (*Buecker and Wysocka, 2012*). Despite overwhelming evidence for multiple enhancers regulating the same gene it is unknown whether they simultaneously interact with the same promoter in a given cell. Here, we use a combination of quantitative live imaging and theoretical modeling to investigate the function of multiple enhancers for the regulation of a common target gene within a single cell type.

The fate map of the adult fly is established by ~1000 enhancers that regulate several hundred patterning genes during the 1-hr interval between two and three hrs after fertilization (*Levine, 2010*; *Nien et al., 2011*). As many as half of these genes contain 'shadow' enhancers with overlapping spatiotemporal activities that are thought to improve the precision and reliability of gene expression

**eLife digest** Only a subset of the genes in a cell is active at any time. Gene activation or 'transcription' is controlled by specific DNA sequences called promoters and enhancers. Promoters are found next to genes and recruit the protein machinery needed to transcribe the gene. Enhancers are located further away from genes and interact with the promoter to increase gene transcription.

Some genes have multiple enhancers with overlapping activities, typically including a primary enhancer and secondary or 'shadow' enhancer that is even more distant. Multiple enhancers are used to control gene transcription in different types of cells. For example, shortly after fertilization around 1000 enhancers in the embryo of the fruit fly *Drosophila* regulate several hundred genes that are important for development. These activities establish which cell type any given cell in the embryo will become. Many of these developmental genes have shadow enhancers, which are thought to make gene activity more precise and reliable. It is not known, however, how several enhancers interact with the same promoter at the same time. For example, do the enhancers' individual effects add together? Or can the combined effects of multiple enhancers be more (or even less) than the sum of their parts?

Bothma, Garcia et al. have now examined how multiple enhancers regulate the activity of three developmental genes (called *hunchback*, *knirps*, and *snail*) in early *Drosophila* embryos. The experiments showed that the individual effects of the *knirps* enhancers add together as one might expect. On the other hand, the *snail* enhancers interfere with each other, which means that their combined effect on transcription is less than the sum of the two individual effects. Furthermore, other experiments revealed that the combined effect of the *hunchback* enhancers depends upon whether another component that is needed for gene activation is in short supply.

To understand these observations, Bothma, Garcia et al. then developed a mathematical model. The model proposes that behavior of enhancers depends upon how strongly they interact with the target promoter. Since 'weak' enhancers do not interact very often, their effects can easily add together. However, the effects of 'strong' enhancers do not add together because they often compete to interact with the promoter. These findings show how multiple enhancers can work in a complex manner to control gene transcription.

(*Hong et al., 2008*; *Frankel et al., 2010*; *Perry et al., 2010*, *2011*; *Lagha et al., 2012*; *Miller et al., 2014*). For example, the *hb* shadow enhancer helps produce a sharp boundary of activation by the Bicoid gradient, while its *snail (sna)* counterpart helps ensure reliable activation under stressful conditions such as high temperatures (*Perry et al., 2010*, *2011*). There is emerging evidence that shadow enhancers are used pervasively in a variety of developmental processes, in both invertebrates and vertebrates (*Lagha et al., 2012*; *Arnold et al., 2013*; *Miller et al., 2014*; *Lam et al., 2015*).

The underlying mechanisms by which two enhancers with extensively overlapping regulatory activities produce coordinated patterns of gene expression are uncertain. It is possible that they augment the levels of gene expression above the minimal thresholds required to execute appropriate cellular processes (*Gregor et al., 2014*). However, there is currently only limited experimental evidence for enhancers acting in an additive fashion (*Arnold et al., 2013*; *Lam et al., 2015*). An alternative view is that shadow enhancers suppress transcriptional noise and help foster uniform expression among the different cells of a population (*Buecker and Wysocka, 2012*). To explore these and other potential mechanisms, we examined the timing and levels of gene activity using bacterial artificial chromosomes (BAC) transgenes containing individual enhancers and combinations of primary and shadow enhancers in the early *Drosophila* embryo.

BAC transgenes containing three key patterning genes, *hb*, *knirps (kni)*, and *sna*, were examined in living precellular embryos. Quantitative analyses suggest that shadow enhancers mediate different mechanisms of transcriptional activity. For *kni*, we observe additive, even super-additive, activities of the primary and shadow enhancer pairs. In contrast, the *hb* enhancers function sub-additively in anterior regions containing saturating levels of the Bicoid activator but function additively in regions where there are diminishing levels of the Bicoid gradient. Strikingly, sub-additive behavior is also observed for sna, in that removal of the proximal enhancer causes a significant increase in gene

expression. These observations suggest that the levels of enhancer activity determine the switch between additive and non-additive behaviors.

Using theoretical modeling, we suggest that these behaviors can be understood in the context of enhancers competing or cooperating for access to the promoter. Weak enhancers work additively due to infrequent interactions with the target promoter, whereas strong enhancers are more likely to impede one another due to frequent associations. Our results highlight the potential of combining quantitative live imaging and modeling in order to dissect the molecular mechanisms responsible for the precision of gene control in development (*Gregor et al., 2014*) and provide a preview into the complex function of multiple enhancers interacting with the same promoter.

## Results

Previous live-imaging studies have relied on simple gene fusions containing a single enhancer attached to a reporter gene with MS2 stem loops inserted in either the 5′ or 3′ UTR (*Garcia et al., 2013*; *Lucas et al., 2013*; *Bothma et al., 2014*). Detection depends on the binding of a maternal mRNA-binding fusion protein (MCP::GFP) expressed throughout the early embryo. In order to examine the interplay between multiple enhancers, we created a series of BAC transgenes containing complete regulatory landscapes (summarized in *Figure 1*). The BAC transgenes contain an MS2-yellow reporter gene in place of the endogenous transcription units (*Figure 1A*). For each locus, *hb*, *kni*, and *sna*, we examined a series of three BAC transgenes: containing both primary and shadow enhancers, as well as derivatives lacking individual enhancers (*Figure 1B,C*). As expected, the BAC transgenes containing both enhancers produce robust expression of the MS2 reporter gene which recapitulate endogenous patterns previously measured using mRNA FISH and immunostaining (*Perry et al., 2010*, *2011*) (*Figure 1D–I*, *Videos 1–3*).

Enhancer 'deletions' were created by substituting native sequences with neutral sequences of similar sizes (see 'Materials and methods'). These substitutions remove most of the critical sequences identified by ChIP-Seq assays (*Harrison et al., 2011*; *Nien et al., 2011*; *Perry et al., 2011*). It is nonetheless possible that critical flanking sequences persist within the transgenes. However, removal of both *kni* enhancers eliminates detectable transcripts in abdominal regions of early embryos (*Figure 1—figure supplement 1*), suggesting that any remaining flanking sequences are insufficient to mediate expression.

Qualitative inspection of the *hb* and *kni* expression videos suggests that removal of either the primary or shadow enhancer does not cause a dramatic alteration in the overall patterns of gene activity. In order to identify more nuanced changes, we quantified the transcriptional activities of the complete series of BAC transgenes (*Figure 2*). The fluorescence intensities of active transcription foci were measured during nuclear cleavage cycles (nc) 13 and 14 at different positions across the anterior–posterior (AP) axis. These intensities were converted into an absolute number of elongating Pol II molecules by calibrating with internal standards (see *Garcia et al., 2013*). Several embryos were analyzed for each time point, and the data were merged to determine the average behavior as a function of AP position and time.

*Hb* expression was examined during the ~15 min interphase of nc 13 when both the primary and shadow enhancers are active, but before the onset of later-acting 'stripe' enhancers during nc 14 (*Perry et al., 2011*, *2012*). We measured the transcriptional activity of all three *hb* BAC transgenes (*Figure 1D,F,H*). Contrary to simple expectations suggested by previous studies (*Arnold et al., 2013*), we find that the two *hb* enhancers do not function in an additive fashion in anterior regions (20–40% egg length, EL) of the embryo (e.g., *Figure 2A,B*). Indeed, the levels produced by the wild-type transgene fall far short of the additive levels predicted by simply summing the levels of expression produced by the transgenes containing either the shadow or primary enhancer alone (*Figure 2A,B*). Moreover, the removal of the shadow enhancer has no effect on the levels of transcription in anterior regions, which is consistent with the original conception of the shadow enhancer as a 'back-up' in the event of stress (*Hong et al., 2008*).

A very different scenario is observed in central regions of the embryo (40–50% EL) where *hb* expression switches from 'on' to 'off' to form a sharp border (*Gregor et al., 2007*). In this region, the wild-type transgene produces significantly higher levels of expression than either of the transgenes driven by a single enhancer. In fact, these levels correspond to the values predicted by simply adding the activities of the single-enhancer transgenes. Thus, the two enhancers transition from *sub-additive* to *additive* behavior in the region of the embryo where there are diminishing levels of the Bicoid

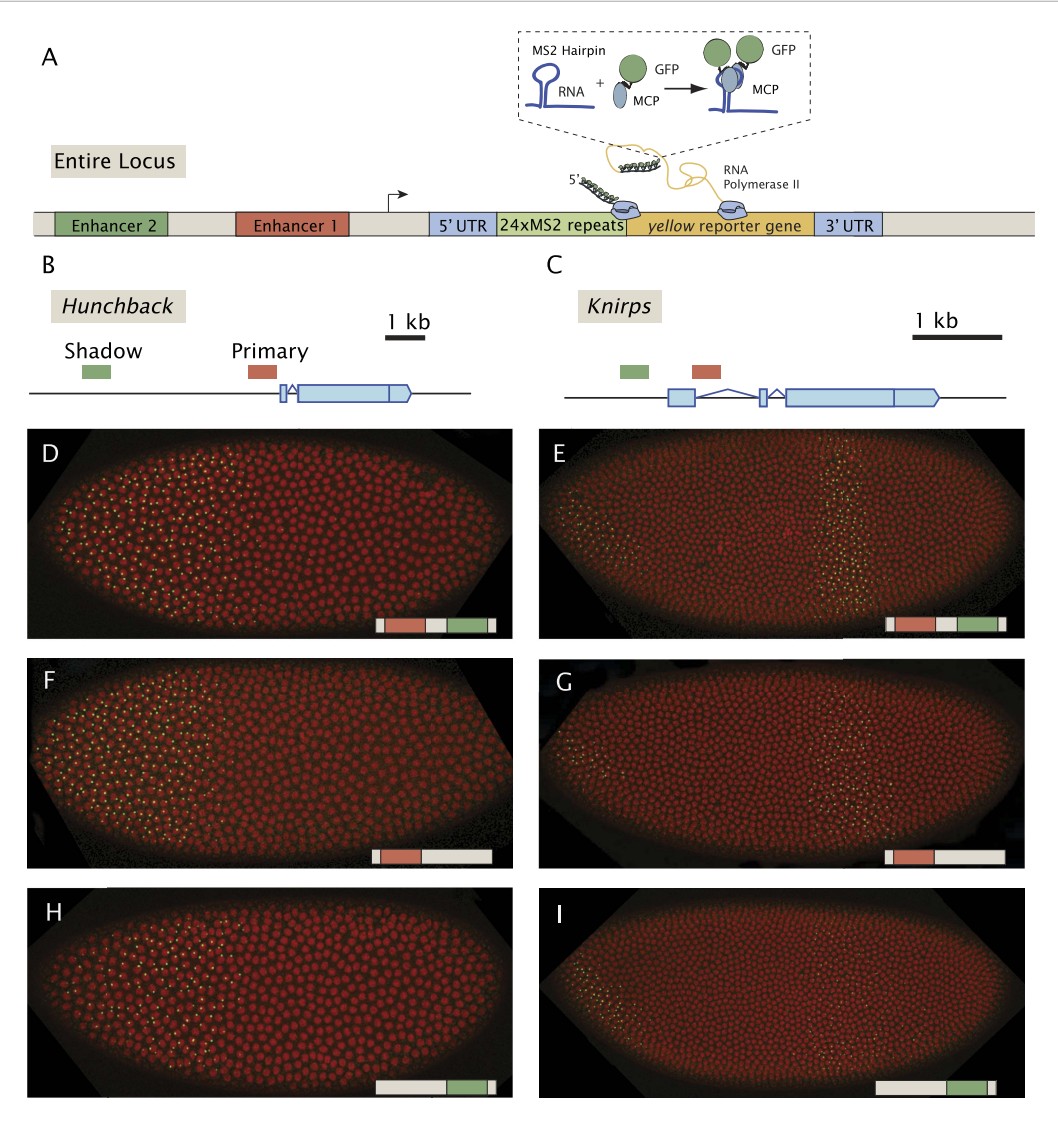

**Figure 1**. Live-imaging of transcriptional activity of *hb* and *kni* loci lacking different enhancers. (**A**) General structure of the reporter constructs. A reporter construct with 24 repeats of the MS2 stem loops and the *yellow* gene was recombined into BACs spanning the *hb* and *kni* loci. The 5′ UTR and 3′ UTR of the endogenous genes were left intact. The MCP::GFP protein that binds to the MS2 stem loops is present in the unfertilized egg and in the early embryo. Gene models of (**B**) the *hb* and (**C**) *kni* loci showing the location of the primary and shadow enhancers (*Perry et al., 2011*). (**D**, **F**, **H**) Snapshots of *Drosophila* embryos expressing different versions of the *hb BAC>MS2* reporter containing different combinations of the two enhancers 10 min into nuclear cleavage cycle 13 (nc13). The colored bar on the bottom right indicates which enhancer was removed. (**E**, **G**, **I**) Snapshots of *Drosophila* embryos expressing different versions of the *kni BAC>*MS2 reporter containing different combinations of the two enhancers in nc14.

The following figure supplement is available for figure 1:

**Figure supplement 1**. *kni* BAC expression lacking both shadow and primary enhancers.

activator gradient. We therefore suggest that the *hb* enhancers function additively only when they are operating below peak capacity (see below).

To further explore the activities of multiple enhancers, we examined the *kni* gene, which is regulated by an intronic enhancer and a distal 5′ enhancer (*Perry et al., 2011*). We focus our analysis on central regions of the abdominal expression pattern since previous studies suggest the occurrence

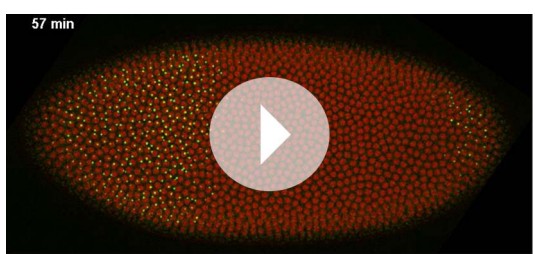

**Video 1.** Dynamics of *hunchback* expression. Maximum projection of *hb* BAC>MS2 transgene from nc10 to gastrulation, MCP::GFP in green and histone in red, anterior to the left and ventral view up. Time elapsed since the start of imaging is indicated in top left. The initial pattern is restricted to the anterior where expression is driven by the primary and shadow enhancers. In late nc13 the central domain enhancer starts to be expressed.

of long-range repressive interactions that establish the borders of the 'stripe' (*Perry et al., 2011*). During early periods of nc 14, the two enhancers function super-additively (*Figure 2C*). That is, the wild-type *kni* BAC transgene produces higher levels of expression than the predicted sum of the two transgenes containing either enhancer alone. During later stages of development, there is a twofold reduction in the expression levels of the endogenous gene, and at this time the two enhancers work in a simple additive manner (*Figure 2D*). Note that the maximum number of elongating polymerase (Pol II) complexes falls short of that seen for *hb* (compare with *Figure 2A,B*).

Understanding the stark difference in the behaviors of the *hb* and *kni* enhancer pairs necessitates measuring the absolute strengths of the different enhancers. Using absolute counts of mRNA molecules (*Little et al., 2013*), we calibrated our live fluorescence intensity traces to determine the average numbers of actively elongating Pol II transcription complexes (*Garcia et al., 2013*). The *kni* transgenes containing single enhancers exhibit as little as fourfold lower levels of expression as compared with the corresponding *hb* transgenes (*Figure 3A,B*). At peak activity, the proximal *hb* enhancer induces ~50 transcribing Pol II complexes across the yellow reporter gene. By contrast, individual *kni* enhancers produce an average of only ~15 elongating Pol II complexes. We propose that the additive and super-additive behaviors of the two *kni* enhancers reflect their inherently 'weaker' activities as compared with the 'stronger' proximal hb enhancer (see 'Discussion'). Note that, despite these differences, the overall output of transcripts and the overall rate of transcript production are essentially identical for all gap genes (*Little et al., 2013*).

To test the proposed anti-correlation between enhancer strength and additivity, we analyzed the expression of *sna*, which is essential for delineating the invaginating mesoderm during gastrulation. *sna* transgenes containing either the proximal or distal enhancer produce peak transcriptional activities of ~40 actively transcribing Pol II complexes across the *yellow* reporter gene, similar to the numbers seen for the proximal *hb* enhancer (*Figure 3C*). Thus, both *sna* enhancers are strong and they exhibit strikingly sub-additive behaviors. In particular, the wild-type transgene displays significantly lower levels of expression than the mutant transgene containing only the shadow enhancer. Thus, strong enhancers not only fail to function additively but also interfere with one another, leading to

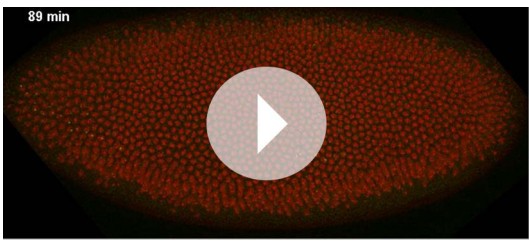

**Video 2.** Dynamics of *knirps* expression. Maximum projection of *kni* BAC>MS2 transgene from nc10 to gastrulation, MCP::GFP in green and histone in red, anterior to the left and ventral view down. Time elapsed since the start of imaging is indicated in top left. The dynamics of the anterior and central parts of the pattern are evident.

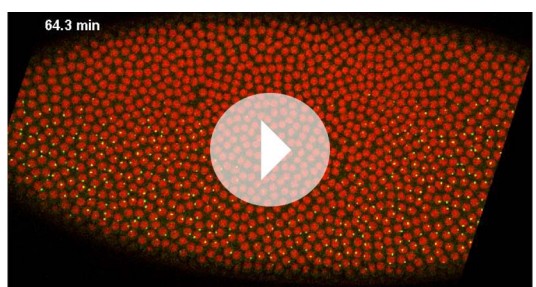

**Video 3.** Dynamics of *snail* expression. Maximum projection of *snail* BAC>MS2 transgene from nc10 to gastrulation, MCP::GFP in green and histone in red, anterior to the left and ventral view up. Time elapsed since the start of imaging is indicated in top left.

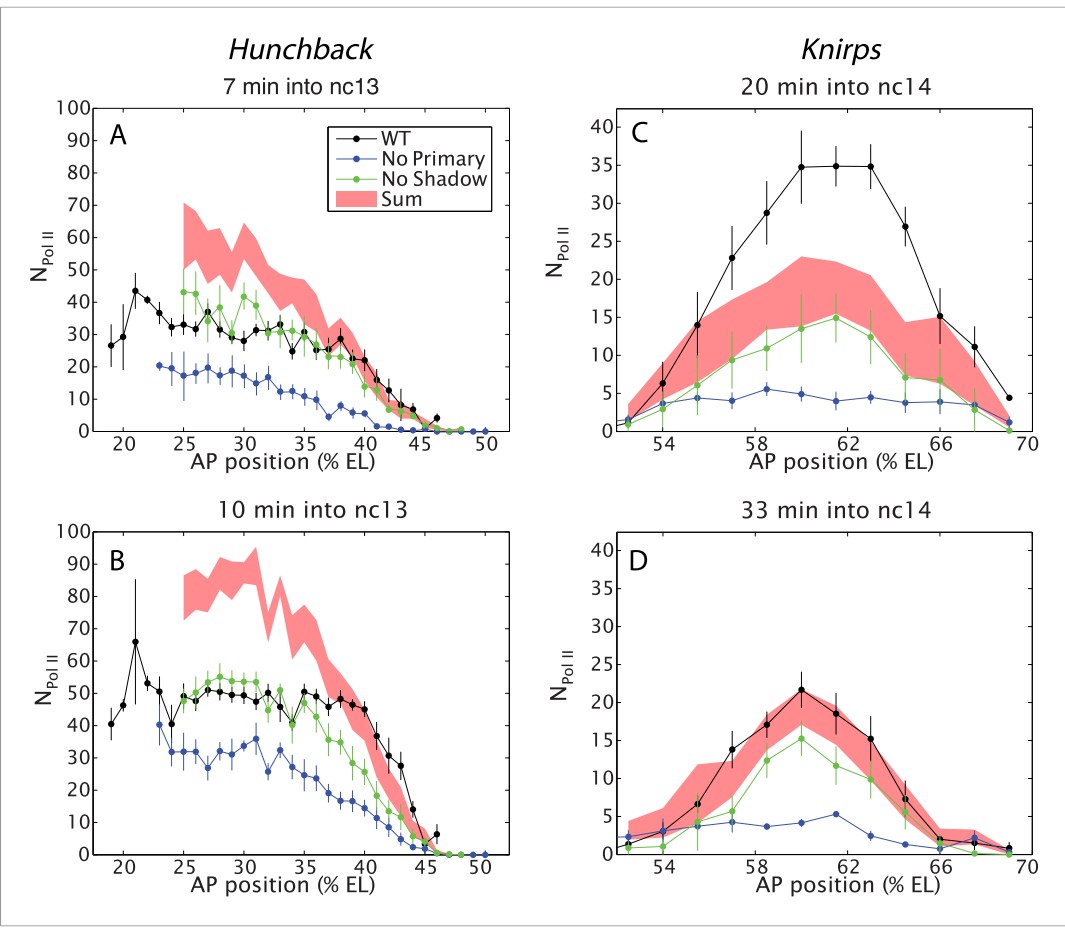

**Figure 2**. Combined effect of multiple enhancers as a function of AP position. (**A**, **B**) Mean number of Pol II molecules transcribing per nucleus ($N_{Pol\ II}$) in the *hb* BAC reporters containing different combinations of enhancers as a function of AP position for two time points in nc13. $N_{Pol\ II}$ is calculated by averaging data from at least three embryos at each AP position. The predicted sum of the individual enhancers is also shown. Note the additivity at the boundary vs the sub-additivity at the core, anterior domain of the pattern. (**C**, **D**) Mean number of Pol II molecules transcribing per nucleus ($N_{Pol\ II}$) in the *kni* BAC reporters in nc14 as a function of AP position. For *kni*, we see super-additive behavior in the beginning of nc14 which then becomes additive later in nc14. The absolute number of transcribing Pol II molecules was estimated following a previous calibration (*Garcia et al., 2013*). Error bars are the standard error of the mean over multiple embryos.

sub-additive expression levels. This observation is also consistent with an earlier study, which suggested that the weaker proximal enhancer attenuates the activities of the stronger distal shadow enhancer (*Dunipace et al., 2011*).

In an effort to understand how multiple enhancers might function additively or sub-additively, we developed a mathematical model for dynamic enhancer–promoter interactions. In this model, a single enhancer interacts with its promoter via a forward rate $k_{on}$ and a backward rate $k_{off}$ (*Figure 4A*). The relative values of the forward and reverse rates determine the strength of the enhancer–promoter interaction by controlling what fraction of time the two are bound. When the enhancer and promoter interact, the promoter is in the ON state and initiates transcription at a rate r. This rate can be interpreted as the efficiency of enhancer-mediated transcriptional initiation upon enhancer–promoter interaction. Hence, the observable rate of mRNA production depends on the interaction strength given by the ratio $k_{on}/k_{off}$, and the efficiency r with which transcription is initiated upon interaction (*Figure 4B*). This scheme can be generalized to include two enhancers (A and B) interacting with the same promoter (*Figure 4C* and see 'Materials and methods' for details of the mathematical analysis). In this model only one enhancer can interact with the promoter at a given time.

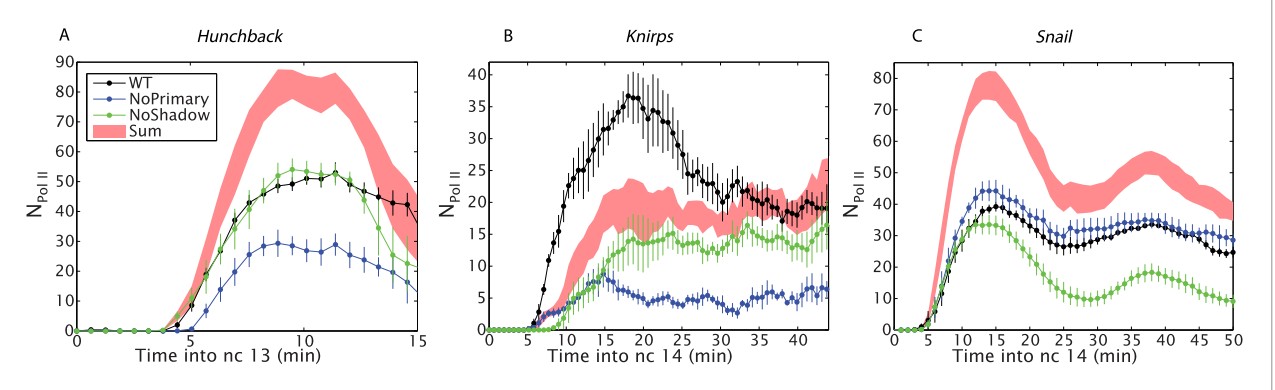

**Figure 3**. Combined effect of multiple enhancers as a function of time. (**A**) Time course of the mean number of Pol II molecules transcribing per nucleus ($N_{Pol\ II}$) for the different *hb* BAC transgenes and sum of individual enhancers at 27% EL for the duration of nc13. (**B**) *kni* BAC transgenes activities and the sum of individual enhancer activity at 60% EL for the first 50 min of nc14. (**C**) *sna* BAC transgenes and the sum of individual enhancer activities averaged over the central mesoderm for the initial 50 min of nc14. Error bars are the standard error of the mean over multiple embryos.

When the individual enhancers interact infrequently with the promoter ($k_{on}/k_{off} << 1$) they are unlikely to attempt to engage the promoter simultaneously. In this regime, the enhancers will work additively, and the rate of mRNA production of enhancers A and B will simply be the sum of the production rate of A and B alone as shown in *Figure 4D*. However, as the strength of promoter–enhancer interactions increases, the combined activity of both enhancers is less than the sum of each individual enhancer. When we model enhancers of different strengths (*Figure 4E*), the amount of mRNA production is reduced as the enhancer with the weaker transcriptional efficiency interacts more frequently with the promoter. This occurs because the two enhancers compete for access to the promoter, effectively inhibiting one another. Thus, weak enhancers might work additively due to infrequent interactions with the target promoter, whereas strong enhancers interfere with one another due to more frequent interactions (see below).

## Discussion

Our quantitative analysis of *hb* and *kni* expression provides seemingly opposing results. For *kni*, we observe additive, sometimes even super-additive, action of the two enhancers within the presumptive abdomen. In contrast, the two *hb* enhancers do not function in an additive fashion in anterior regions but are additive only in central regions where expression abruptly switches from 'on' to 'off'. We propose that 'weak' enhancers function additively or even super-additively, whereas 'strong' enhancers can impede one another (*Figure 5*).

Additional support for this view is provided by the analysis of *sna*. We found that the removal of the proximal enhancer significantly augments expression, consistent with the occurrence of enhancer interference within the native locus. It is also conceivable that a single strong enhancer (e.g., *hb* proximal or *sna* distal) already mediates maximum binding and release of Pol II at the promoter, and additional enhancers are therefore unable to increase the levels of expression. However, the increase in the levels of *sna* expression upon removal of the primary enhancer is inconsistent with this explanation. Perhaps, the proximity of the proximal enhancer to the *sna* promoter gives it a 'topological advantage' in blocking access of the distal enhancer (*Dunipace et al., 2011*). The proximal enhancer might mediate less efficient transcription than the distal enhancer and thereby reduce the overall levels of expression (see *Figure 4E*). We do not believe that this proposed difference is due to differential rates of Pol II elongation since published (*Garcia et al., 2013*) and preliminary studies suggest that different enhancers and promoters lead to similar elongation rates (~2 kb/min; T Fuyaka and M Levine, unpublished results). A nonexclusive alternative possibility is that deletion of the proximal enhancer removes associated *sna* repression elements (*MacArthur et al., 2009*), thereby augmenting the efficiency of the distal enhancer.

A minimal model of enhancer–promoter associations provides insights into potential mechanisms. In the parameter regime where such interactions are infrequent the two enhancers display additive

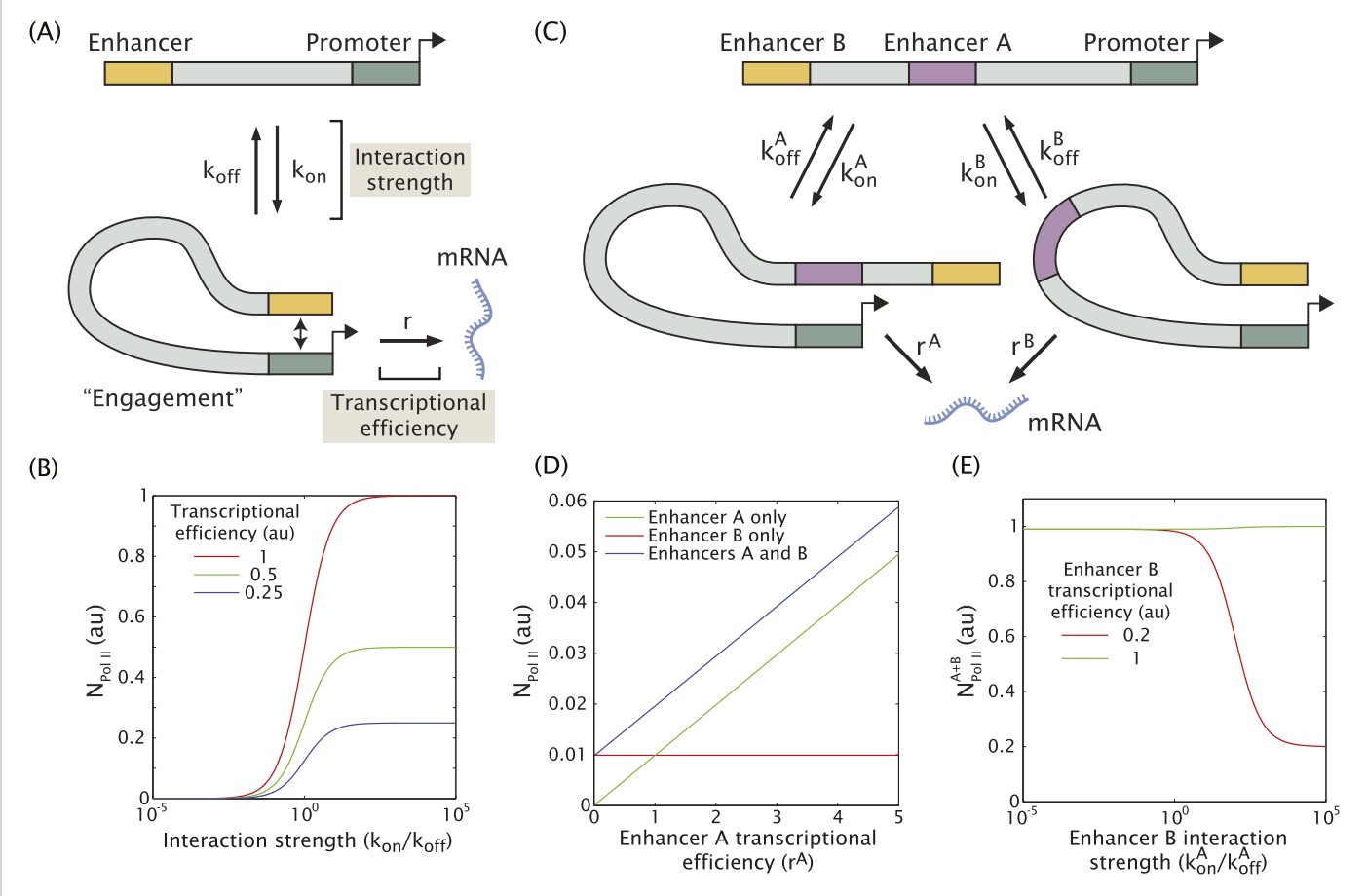

**Figure 4**. Model of enhancer–promoter interactions and its predictions for mRNA production. (**A**) Minimal model of one enhancer engaging a promoter. $k_{on}$ and $k_{off}$ are the rates of promoter engagement and disengagement, respectively, and determine the interaction strength. r is the rate of mRNA production when the promoter is engaged and is a measure of the transcriptional efficiency. The mean number of Pol II molecules transcribing per nucleus ($N_{Pol\ II}$) is proportional to the rate of mRNA production. (**B**) As the interaction strength of a single enhancer is increased, the amount of mRNA produced increases up to a maximum value dictated by the transcriptional efficiency. (**C**) The model in (**A**) can be generalized to allow for multiple enhancers interacting with the same promoter. (**D**) In the regime where the interaction strength of both promoters is weak ($k_{on}/k_{off}$ = 0.01), the amount of mRNA produced by having both **A** and **B** is simply the sum of the individual contributions of **A** and **B**, (r = 1). (**E**) In the regime where the interaction strength is large, the combined activity of both enhancers can be significantly less than the sum the individual enhancers. A less efficient enhancer A ($r^A$ = 0.2 au) can interfere with the more efficient enhancer B ($r^B$ = 1 au) such that their combined activity is significantly less than the sum of the activities of individual enhancers.

behavior. However, in the regime of frequent interactions, enhancers compete for access to the promoter resulting in sub-additive behavior. Enhancer–promoter interaction parameters are likely to vary not only between different enhancers but also as the input patterns are modulated in time and space during development (*Rushlow and Shvartsman, 2012*; *Kok and Arnosti, 2015*).

This simple model explains the switch from sub-additive to additive enhancer activities for *hb* and *sna*. However, in order to explain the super-additive behavior of the *kni* enhancers, it would be necessary to incorporate an additional state in the model, whereby both enhancers form an active complex with the same target promoter. Such a complex would have a more potent ability to initiate transcription than individual enhancer–promoter interactions.

In summary, we propose that enhancers operating at reduced activities ('weak enhancers') can function in an additive manner due to relatively infrequent interactions with their target promoters. In contrast, 'strong' enhancers might function sub-additively due to competition for the promoter (*Figure 4E*). For *hb*, this switch between competitive and additive behavior occurs as the levels of Bicoid activator diminish in central regions where the posterior border of the anterior Hb domain is

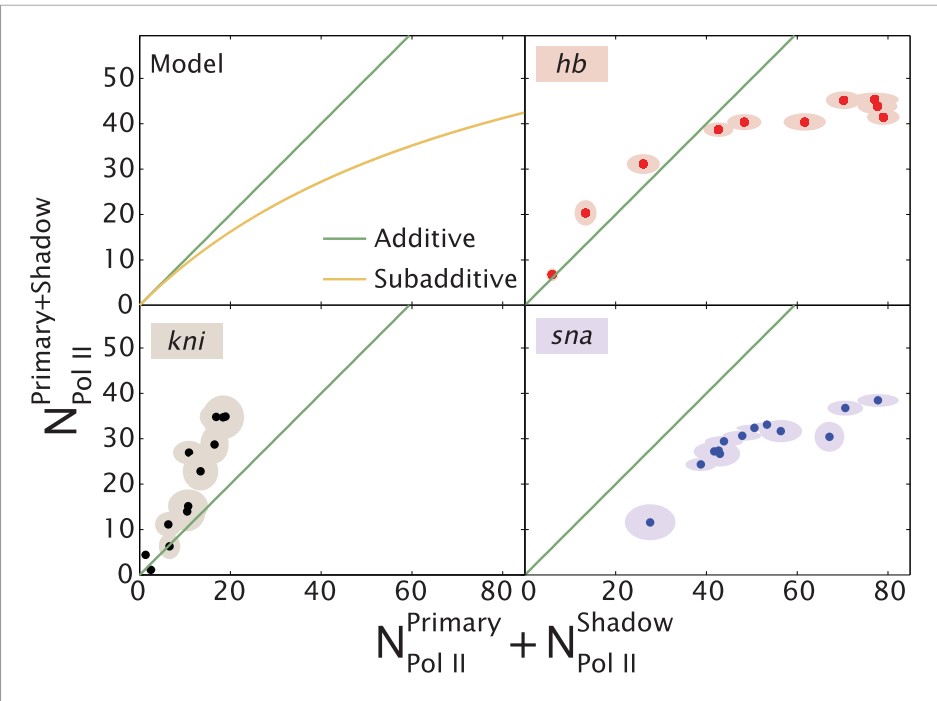

**Figure 5**. Theoretical expectation and experimental results showing different regimes of combined enhancer action. (Upper left) Theoretical predictions (yellow) illustrating how the rate of mRNA production from both enhancers, $N_{Pol\ II}^{Primary+Shadow}$, varies with the sum of the activity of the individual enhancers, $N_{Pol\ II}^{Primary} + N_{Pol\ II}^{Shadow}$ (yellow). Mean number of Pol II molecules transcribing per nucleus ($N_{Pol\ II}$) is proportional to the rate of mRNA production. The green line shows perfect additivity for comparison. The model predicts additive behavior ($N_{Pol\ II}^{Primary+Shadow} \approx N_{Pol\ II}^{Primary} + N_{Pol\ II}^{Shadow}$) when the rate of production is low and sub-additive behavior ($N_{Pol\ II}^{Primary+Shadow} < N_{Pol\ II}^{Primary} + N_{Pol\ II}^{Shadow}$) as the production rate increases. As the interaction strength of individual enhancers increases so does the rate of mRNA production, but the combined activity of both enhancers becomes sub-additive. (Upper right, lower left, lower right) Transcriptional activity of intact loci vs the sum of activities of individual enhancers for *hb*, *kni*, and *sna* at different times. A green line has been drawn in to indicate where $N_{Pol\ II}^{Primary+Shadow}$ is equal to $N_{Pol\ II}^{Primary} + N_{Pol\ II}^{Shadow}$. For *hb* and *kni*, the plots show data taken at different AP positions at 10 min into nc 13 and 20 min into nc 14, respectively, while for *sna* the datapoints were at different times. Ellipses indicate standard error of the mean.

formed. Similarly, stress might reduce the performance of the *sna* enhancers to foster additive behavior under unfavorable conditions such as increases in temperature (*Perry et al., 2010*). Our study highlights the complexity of multiple enhancers in the regulation of gene expression. They need not function in a simple additive manner, and consequently, their value may be revealed only when their activities are compromised.

## Materials and methods

### Cloning and recombineering

In brief, BAC clones that map to the region of interest were identified from end-sequenced BAC libraries which can be viewed on a browser at http://pacmanfly.org, and ordered from BacPac Resources (http://bacpac.chori.org/) (*Venken et al., 2009*). These BACs arrive already cloned into a vector containing an attB sequence for targeted integration, mini-white cassette, chloramphenicol resistance and are in the inducible copy number strain EPI300, Epicentre Biotechnologies (Madison, WI). The following CHORI BACs were used as a starting point: *sna* BAC (CH322-18I14-1), *hb* BAC (BAC CH322-55J23), *kni* BAC (CH322-21P08).

BACs requiring modification were first transformed into the recombineering strain SW102, which was obtained from NCI-Frederick Biological Resources Branch. Cultures containing specific BACs were grown overnight and recombination functions were induced as described (*Perry et al., 2010*). The induced bacteria were electroporated with targeting constructs that were prepared previously by PCR amplification. Targeting constructs were made using a pair of 90 base pair long oligonucleotides. These contained 25 base pairs specific to the region being amplified that was to be swapped into the BAC, and an additional 65 base pairs of sequence homologous to the target BAC flanking the region to be replaced. The homologous regions, or 'homology arms', target the amplified sequence to the region of interest for recombination. After electroporation and a 1 hr recovery period in 2XYT broth, bacteria were plated in a dilution series on LB plates with the appropriate antibiotic for overnight incubation at 30°C. Individual resulting colonies were screened by PCR for appropriate recombination at both homology arm locations. Confirmed positive recombinant colonies were transformed back into EPI300 cells (Epicentre Biotechnologies) and reconfirmed by antibiotic marker selection and PCR; PCR products were sequenced for final confirmation.

Oligonucleotides for amplification to make homology arm constructs (90 base pairs in length) were from Integrated DNA Technologies; shorter primers for colony screening PCR were from ELIM Biopharmaceuticals. Restriction enzymes were from New England Biopharmaceuticals. Qiagen products were used to isolate plasmid DNAs, gel-purify DNA fragments, and purify PCR products. Qiagen taq polymerase was used in colony PCR screening; Invitrogen Platinum pfx was used to amplify targeting constructs.

The first step in the modification was to replace the endogenous coding sequence of *sna*, *hb*, and *kni* genes with that of the yellow-kanamycin reporter gene. The yellow-kanamycin fragment was swapped into the place of the endogenous gene at the ATG start codon at the 5′ end, leaving the 5′ UTR intact. The endogenous 3′ UTR was also left fully intact. In most cases, the different enhancers were replaced with an ampicillin resistance cassette which was PCR amplified from pBluescript. In the case of kni, one of the enhancers is in the intron of the transcribed region and so we replaced enhancer with a fragment of lambda phage DNA using galK positive and negative selection. The next step required the insertion the MS2 stem loop sequences. Copies of the MS2 stem loops were extracted from plasmid pCR4-24XMS2SL-stable (Addgene 31865) and were PCR amplified with primers with appropriate homology sequences.

## BAC preparation for microinjection and phiC31-mediated integration

BACs were induced to high copy number using Epicentre BAC autoinduction solution, according to supplier's instructions, and grown overnight for 16–18 hr at 37°C. DNA was prepared for micro-injection using the Invitrogen PureLink HiPure miniprep kit by following manufacturer instructions with described modifications for BACs and cosmids. DNA was diluted to a final concentration of ∼300–400 ng/μl and 1× injection buffer. At least 200 embryos were injected per construct by BestGene Inc. (Chino Hills, CA). The transgenes were integrated into the following landing sites: BDSC 9723, BDSC 9750, and BDSC 24749. *Hb* lacking shadow and *kni* lacking primary were integrated into 9750 and 24,749, respectively, while all other transgenes were integrated into 9723.

### Live imaging sample preparation and data acquisition

Female virgins of line yw; Histone-RFP;MCP-NoNLS-GFP (*Garcia et al., 2013*) were crossed with males of each reporter line. Collected embryos were dechorinated using bleach and mounted between a semipermeable membrane (Biofolie, In Vitro Systems & Services) and a coverslip (1.5, 18 mm × 18 mm) and embedded in Halocarbon 27 oil (Sigma). The flattening of the embryos makes it possible to image a larger number of nuclei in the same focal plane without causing significant changes in early development processes (*Di Talia and Wieschaus, 2012*).

Embryos were either imaged using a custom-built two-photon microscope (*Liu et al., 2013*) and a Zeiss LSM 780 confocal microscope. Imaging conditions on the two-photon microscope were as described in *Garcia et al. (2013)*. The average laser power at the specimen was 10 mW, the pixel size was set to 220 nm and a single image consisted of 512 × 256 pixels. At each time point, a stack of 10 images separated by 1 μm was acquired resulting in a final time resolution of 37 s. Confocal imaging was performed using a Plan-Apochromat 40×/1.4NA oil immersion objective. The MCP-GFP and Histone-RFP were excited with a laser wavelength of 488 nm and 561 nm, respectively. Fluorescence was detected with two separate photomultiplier tubes using the Zeiss QUASAR

detection unit (gallium-arsenide-phosphide photomultiplier was used for the GFP signal while the conventional detector was used for the RFP). Pixel size is 198 nm and images were captured at 512 × 512 pixel resolution with the pinhole set to a diameter of 116 µm. At each time point, a stack of 22 images separated by 0.5 µm was captured, spanning the nuclear layer. The final time resolution is 32 s.

## Live imaging data analysis

Analysis was performed as described (*Garcia et al., 2013*) and full code can be downloaded from https://github.com/PrincetonUniversity/FlyRNAQuant. Histone-RFP slices were maximum projected for each time point. Nuclei were segmented using an object detection approach based on the Laplacian of Gaussian filter kernel. The segmented nuclei were then segmented and tracked over multiple nuclear cycles. Spots are detected in 3D and assigned to their respectively closest nucleus. When multiple spots are detected in the vicinity of a nucleus only the brightest one is kept. Spot intensity determination necessitates an estimate of the local fluorescent background for each particle. A 2D Gaussian fit to the peak plane of each particle column determines an offset, which is used as background estimator. The intensity is calculated by integrating the particle fluorescence over a circle with a radius of 6 pixels and then subtracting the estimated background. The imaging error is dominated by the error made in the fluorescent background estimation (*Garcia et al., 2013*).

It is possible to measure the average fluorescence per polymerase molecule for the *hunchback enhancer > MS2* transgene with 24 MS2 repeats (*Garcia et al., 2013*). The quantitative imaging for the BAC transgenes was conducted under the exact same imaging conditions on the same microscope. The BAC transgenes also possess 24 MS2 repeats. However, the specific sequence of the stem loops is slightly different as these repeats have been further optimized to facilitate molecular biology work with them (*Hocine et al., 2013*). Assuming that the MS2 sites are similarly saturated with MCP::GFP protein in both cases we can then use the average fluorescence per polymerase molecule calculated for the *hunchback>MS2* transgene to calibrate the BAC fluorescent traces in terms of the absolute number of transcribing polymerases per fluorescent spot.

## Mathematical modeling

We propose a general scheme for enhancer promoter interactions which makes it possible to model the effect of having multiple enhancers activating a single promoter. Here, the enhancer and promoter engage and disengage with one another with characteristic rate constants $k_{on}$ and $k_{off}$, respectively (see *Figure 4A*). The ratio between $k_{on}$ and $k_{off}$ determines the strength of the promoter-enhancer interaction. The system can be found in two states. First, the promoter can be unoccupied. We denote the occupancy of this state with [Enhancer]. Second, the promoter and enhancer can be engaged with an occupancy [Enhancer·Promoter]. Following the reaction scheme shown in *Figure 4A*, the temporal evolution of the occupancy of the state where the enhancer and promoter are engaged is given by

$$\frac{d[\text{Enhancer} \cdot \text{Promoter}]}{dt} = k_{on}[\text{Promoter}] - k_{off}[\text{Enhancer} \cdot \text{Promoter}]. \qquad (1)$$

While the enhancer is engaged with the promoter it is capable of producing mRNA at a rate r, which we will call the transcriptional efficiency. Hence the rate of mRNA production is given by

$$\frac{d\text{mRNA}}{dt} = r[\text{Enhancer} \cdot \text{Promoter}]. \qquad (2)$$

Since the promoter can only be in two states, engaged or not engaged by the enhancer, the occupancy of these two states is constrained by

$$[\text{Enhancer} \cdot \text{Promoter}] + [\text{Promoter}] = 1. \qquad (3)$$

Assuming steady state in the temporal evolution of the state occupancies results in a rate of mRNA production of

$$\frac{d\text{mRNA}}{dt} = \frac{k_{on}}{k_{on} + k_{off}} * r. \qquad (4)$$

In *Figure 4B* we plot this rate as a function of the interaction strength for different values of enhancer efficiency.

When considering two enhancers that can interact independently with the promoter (*Figure 4C*) we need to calculate the occupancy of the state accounting for the promoter interacting with enhancer A, [Enhancer^A·Promoter], and with enhancer B, [Enhancer^B·Promoter]. The temporal evolution of these occupancies is given by

$$\frac{d\left[Enhancer^A \cdot Promoter\right]}{dt} = k_{on}^A[Promoter] - k_{off}^A\left[Enhancer^A \cdot Promoter\right], \qquad (5)$$

and

$$\frac{d\left[Enhancer^B \cdot Promoter\right]}{dt} = k_{on}^B[Promoter] - k_{off}^B\left[Enhancer^B \cdot Promoter\right]. \qquad (6)$$

In this case the rate of mRNA production is given by

$$\frac{dmRNA}{dt} = r^A\left[Enhancer^A \cdot Promoter\right] + r^B\left[Enhancer^B \cdot Promoter\right], \qquad (7)$$

where the different promoter occupancy states are constrained by

$$\left[Enhancer^A \cdot Promoter\right] + \left[Enhancer^B \cdot Promoter\right] + [Promoter] = 1. \qquad (8)$$

In analogy to the single-enhancer case we now assume steady state for the temporal evolution of the occupancies described by *Equations 5, 6* and use the constraint given by *Equation 7* to solve for [Enhancer^A·Promoter] and [Enhancer^B·Promoter]. Finally we replace these results into *Equation 8* leading to

$$\frac{dmRNA}{dt} = \frac{r^A k_{on}^A k_{off}^B + r^B k_{on}^B k_{off}^A}{k_{on}^B k_{off}^A + k_{on}^A k_{off}^B + k_{off}^A k_{off}^B}. \qquad (9)$$

Using this equation we determine the rate of mRNA production as a function of the strength (i.e., r) and the efficiency (i.e., $k_{on}/k_{off}$) of enhancers A and B. These calculations are used to generate the plots shown in *Figure 4D,E*.

## Acknowledgements

We thank Y Yamazaki and E Esposito for help with molecular biology and A Sanchez for discussions on theoretical modeling.

## Additional information

### Funding

| Funder | Grant reference | Author |
| --- | --- | --- |
| National Institutes of Health (NIH) | GM34431 | Michael Levine |
| National Institutes of Health (NIH) | GM071508 | Thomas Gregor |
| National Institutes of Health (NIH) | GM097275 | Thomas Gregor |
| Burroughs Wellcome Fund | 101880 | Hernan G Garcia |
| Physics Department, Princeton University | Dicke Fellowship | Hernan G Garcia |
| Searle Scholar Award | 10-SSP-274 | Thomas Gregor |

The funders had no role in study design, data collection and interpretation, or the decision to submit the work for publication.

### Author contributions

JPB, HGG, Conception and design, Acquisition of data, Analysis and interpretation of data, Drafting or revising the article, Contributed unpublished essential data or reagents; TG, ML, Conception and

design, Analysis and interpretation of data, Drafting or revising the article; SN, Acquisition of data, Analysis and interpretation of data, Contributed unpublished essential data or reagents; MWP, Conception and design, Contributed unpublished essential data or reagents

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
