## [Decision Letter]

Thank you for submitting your work entitled “Enhancer synergy and interference in living *Drosophila* embryos” for peer review at *eLife*. Your submission has been favorably evaluated by Jim Kadonaga (Senior editor), Robb Krumlauf (Reviewing editor), and three reviewers.

The reviewers have discussed the reviews with one another and the Reviewing editor has drafted this decision to help you prepare a revised submission.

The general consensus of the reviewers is that this is a strong paper, detailing high quality work with wide general relevance and import to current questions in the field of gene regulation. The main conclusion, that enhancers can work together in a variety of ways and that these differences are linked to important factors such as enhancer strength, is a significant one that will likely have an impact on the field. The state-of-the-art technique used here to visualize transcription live in vivo, represents a major advance for the quantitative study of enhancer function and gene expression.

No major experimental revisions are required but comments and concerns were raised by all three reviewers primarily in two areas: (1) relevant considerations that are not discussed in the text, and (2) limitations of the very simple theoretical model.

There is real enthusiasm for seeing a revised manuscript that incorporates or rebuts the issues and specific points noted below. The collective view is that this will significantly enhance the general interest of the manuscript.

The points to consider in revision are:

1) The authors' interpretations depend on the assumption that 100% of the (relevant) regulatory activity at each gene locus is contained within the identified enhancers, and that flanking sequences left in the loci after deletion of an enhancer have no impact on gene expression. However, it is possible that sequences that flank enhancer A could interact positively with enhancer B when A is deleted, resulting in a “super-additive” outcome. Minimal versions of enhancers are often weaker than longer versions, so it is possible that not all of the 'primary' or 'shadow' enhancer activity has been removed by the deletions. I think that the authors' assumption is a reasonable one, but it might be safer to acknowledge that when they refer to a transgene containing the primary enhancer “alone” (Results, fifth paragraph), the deletion transgene actually contains much more non-coding sequence than that.

2) The idea that enhancer additivity is a function of enhancer strength is a very interesting and appealing one, and it is central to the paper. Using the existing data, is there a way to plot those two measurements against one another to illustrate this idea more directly?

3) I understand the theoretical model (Figure 4) as it is described in the text, but the authors make a point of saying that it explains enhancer interference, where “the combined activity of both enhancers is less than the sum of both enhancers” (at the end of Results), and I don't see any evidence of enhancer interference in the graph. Shouldn't the red line (2 enhancers) dip below the green line (1 enhancer), and shouldn't the blue line reach an expression ratio lower than 1?

4) On a closely related note, it isn't obvious to me how the model described here could ever drive the expression ratio below 1 by increasing enhancer-promoter engagement time, unless one enhancer were better at engaging the promoter but also worse at activating transcription when engaged. Wouldn't another state, such as non-productive enhancer-promoter interaction, need to be added to the model to achieve interference?

5) In 2011, Perry et al. showed that the knirps intronic enhancer drives an inappropriately wide expression pattern, one that is “normalized” when coupled with the proximal enhancer. This result suggests that the latter element has long-range repression type effects (or acts on the basal promoter in nuclei normally not expressing knirps to silence would-be inputs from the intron element). This data indicates that the authors' modeling of the enhancers as small activating units does not really conform to what we know about these elements. The authors do not discuss this prior finding, but presumably they have seen similar effects in their two photon system? The authors should test how a model involving such negative regulatory inputs would change their predictions. Admittedly this model would be more complex than the simple on/off model presented in this version of the manuscript, but we already know that this simpler presentation is insufficient.

6) In 2009, MacArthur and colleagues measured transcription factor occupancy in the embryo. In that study, these researchers found that the Snail transcriptional repressor directly binds to the Snail regulatory elements that are tested here. The expression of Snail would thus induce some sort of negative feedback, if the binding measured in that study is indeed active. In light of the findings from this manuscript, where the primary enhancer by itself is weaker, and the shadow enhancer by itself is able to drive gene expression at about wild-type levels, it appears that Snail self-inhibition may be compromising promoter activity to reduce the overall output. Previous studies of the related short-range repressor Knirps have shown that this class of repressor may induce local, heterochromatized-like chromatin status (Li and Arnosti, 2011), thus if Snail is doing the same thing, the activity of this proximal element should be considered as a mix of active “silencer”-like activity with “enhancer”-like action in the classic sense. A similar local silencing action by parts of the knirps proximal element would explain the Perry results.

7) The authors should address whether differences in elongation rates (which may be unlinked to promoter firing) would affect their estimations of overall enhancer activity. The genes tested here are very similar, but not identical – the lambda DNA filler for knirps or the unique 5' and 3' UTRs may have differential elongation potential. The overall conclusions about enhancer (non) additivity would still hold in any event.

8) Authors state that hb enhancers do not function in additive fashion in anterior regions of the embryo (are subadditive). It is true, but by looking at the data in Figures 2 and 3 would say that the transcriptional output of the shadow enhancer deletion overlaps pretty well with the wt, suggesting that hb shadow enhancer does not provide much of a contribution to transcription, at least in the anterior part of the embryo, which perhaps could be stated explicitly.

9) Clearly the theoretical model does not account for the superadditive enhancer behavior, as seen for kni enhancers (which authors acknowledge in the discussion). Nonetheless, existence of the superadditive behavior indicates that a central assumption of the model – that a single enhancer engages the promoter at the time – is probably incorrect at least at a subset of loci. Could authors incorporate an additional state to their model accounting for the possibility of multiple enhancer interactions? How would that affect predictions of their model? What do authors think may determine competitive vs cooperative enhancer engagement with the promoter?

---

## [Author Response]

*1) The authors' interpretations depend on the assumption that 100% of the (relevant) regulatory activity at each gene locus is contained within the identified enhancers, and that flanking sequences left in the loci after deletion of an enhancer have no impact on gene expression. However, it is possible that sequences that flank enhancer A could interact positively with enhancer B when A is deleted, resulting in a* “*super-additive*” *outcome. Minimal versions of enhancers are often weaker than longer versions, so it is possible that not all of the 'primary' or 'shadow' enhancer activity has been removed by the deletions. I think that the authors' assumption is a reasonable one, but it might be safer to acknowledge that when they refer to a transgene containing the primary enhancer* “*alone*” *(Results, fifth paragraph), the deletion transgene actually contains much more non-coding sequence than that*.

Concerns were raised about the persistence of critical flanking sequences upon deletion of core enhancers. We have added a new paragraph in the Results to address this important issue. The enhancer deletions are actually sequence substitutions, as we now describe in the revised Results. Every effort was made to replace all of the critical binding sites identified in previous ChIP-Seq assays. We nonetheless concede the possibility that critical elements are left intact. However, new data are included to show that any such sequences are not sufficient to mediate activation. For example, “deletions” of both the 5’ and intronic enhancers within the kni locus completely abolish expression of the corresponding transgene in the presumptive abdomen (see Figure 1—figure supplement 1).

*2) The idea that enhancer additivity is a function of enhancer strength is a very interesting and appealing one, and it is central to the paper. Using the existing data, is there a way to plot those two measurements against one another to illustrate this idea more directly*?

We were asked to revise Figure 5 to provide a clearer representation of the anti-correlation between enhancer strength and additivity. The revised Figure 5 shows how the combined activity of the two enhancers varies as a function of the sum of their individual activities. For hb, when the combined activities are low (left hand side of the x axis) the combined activity is additive. However as the strength of the individual activities increases (right hand side of the x axis) the combined activity becomes sub-additive.

*3) I understand the theoretical model (*Figure 4*) as it is described in the text, but the authors make a point of saying that it explains enhancer interference, where* “*the combined activity of both enhancers is less than the sum of both enhancers*” *(at the end of Results), and I don't see any evidence of enhancer interference in the graph. Shouldn't the red line (2 enhancers) dip below the green line (1 enhancer), and shouldn't the blue line reach an expression ratio lower than 1*?

We are asked to provide a clearer depiction of enhancer interference. Yes, we agree that the original version of Figure 4 was unclear. We now present a revised version that more clearly illustrates enhancer interference. Moreover, Figure 5 was also revised to show how two enhancers are less than the sum of their individual contributions in the regime of high rates of mRNA production.

*4) On a closely related note, it isn't obvious to me how the model described here could ever drive the expression ratio below 1 by increasing enhancer-promoter engagement time, unless one enhancer were better at engaging the promoter but also worse at activating transcription when engaged. Wouldn't another state, such as non-productive enhancer-promoter interaction, need to be added to the model to achieve interference*?

We were asked to consider the possibility of non-productive or weakly productive enhancer-promoter interactions to account for the interference phenomenon seen in the *snail* locus. Yes, this is an excellent point. The text has been revised to include this possibility. Moreover, we have plotted how mRNAs levels change with enhancer-promoter engagement times and different enhancer strengths. When one enhancer is weak and the other strong there is decreased mRNA synthesis as the frequency of engagement by the weak enhancer goes up.

We agree with the referee that this is probably the case for snail, since the shadow enhancer is stronger than the proximal enhancer. We state this clearly in the revised text.

*5) In 2011, Perry et al. showed that the knirps intronic enhancer drives an inappropriately wide expression pattern, one that is* “*normalized*” *when coupled with the proximal enhancer. This result suggests that the latter element has long-range repression type effects (or acts on the basal promoter in nuclei normally not expressing knirps to silence would-be inputs from the intron element). This data indicates that the authors' modeling of the enhancers as small activating units does not really conform to what we know about these elements. The authors do not discuss this prior finding, but presumably they have seen similar effects in their two photon system? The authors should test how a model involving such negative regulatory inputs would change their predictions. Admittedly this model would be more complex than the simple on/off model presented in this version of the manuscript, but we already know that this simpler presentation is insufficient*.

We were asked to address the occurrence of long-range repression within the knirps locus. We have revised the text to indicate that this type of repression has been previously documented. We also emphasize that the enhancer analysis is focused on central regions of the knirps expression pattern, outside the limits of the repressive interactions that help establish the sharp borders of the expression “stripe”.

*6) In 2009, MacArthur and colleagues measured transcription factor occupancy in the embryo. In that study, these researchers found that the Snail transcriptional repressor directly binds to the Snail regulatory elements that are tested here. The expression of Snail would thus induce some sort of negative feedback, if the binding measured in that study is indeed active. In light of the findings from this manuscript, where the primary enhancer by itself is weaker, and the shadow enhancer by itself is able to drive gene expression at about wild-type levels, it appears that Snail self-inhibition may be compromising promoter activity to reduce the overall output. Previous studies of the related short-range repressor Knirps have shown that this class of repressor may induce local, heterochromatized-like chromatin status (Li and Arnosti, 2011), thus if Snail is doing the same thing, the activity of this proximal element should be considered as a mix of active* “*silencer*”*-like activity with* “*enhancer*”*-like action in the classic sense. A similar local silencing action by parts of the knirps proximal element would explain the Perry results*.

We were asked to consider the possibility that Snail repression elements located within or near the proximal enhancer might attenuate gene expression. This is a valid point and we have revised the text to consider this possibility and now include the MacArthur et al. 2009 citation.

*7) The authors should address whether differences in elongation rates (which may be unlinked to promoter firing) would affect their estimations of overall enhancer activity. The genes tested here are very similar, but not identical – the lambda DNA filler for knirps or the unique 5' and 3' UTRs may have differential elongation potential. The overall conclusions about enhancer (non) additivity would still hold in any event*.

We were asked whether different Pol II elongation rates might complicate our interpretations. The text has been revised to address this possibility, and we also cite recent and unpublished evidence that many different enhancer-promoter combinations produce the same rate of elongation, approximately 30 nt/s.

*8) Authors state that hb enhancers do not function in additive fashion in anterior regions of the embryo (are subadditive). It is true, but by looking at the data in*
Figures 2 and 3
*would say that the transcriptional output of the shadow enhancer deletion overlaps pretty well with the wt, suggesting that hb shadow enhancer does not provide much of a contribution to transcription, at least in the anterior part of the embryo, which perhaps could be stated explicitly*.

We were asked to clarify that removal of the *hb* shadow enhancer has no effect on the levels of transcription in anterior regions of the embryo. This is now done in the revised manuscript.

*9) Clearly the theoretical model does not account for the superadditive enhancer behavior, as seen for kni enhancers (which authors acknowledge in the discussion). Nonetheless, existence of the superadditive behavior indicates that a central assumption of the model – that a single enhancer engages the promoter at the time – is probably incorrect at least at a subset of loci. Could authors incorporate an additional state to their model accounting for the possibility of multiple enhancer interactions? How would that affect predictions of their model? What do authors think may determine competitive vs cooperative enhancer engagement with the promoter*?

It was noted that our model fails to explain the super-additive behavior of the knirps enhancers during early stages of nc 14. We really do not have a good explanation for super-additivity. During the later stages of nc 14, the two enhancers function in a simple additive fashion, and this behavior is captured by our model. At earlier stages it would appear that the interaction of one enhancer facilitates the interaction of the other enhancer. This possibility is clearly stated in the Discussion. However, we do not know enough about this type of enhancer cooperativity to formulate a reasonable model. Any such effort would be wildly speculative.